# AthlyticsMind: A Tailored LLM-based Conversational Agent to Mitigate Stigma and Support Elite Athlete Mental Health

## Abstract

Elite athletes, often perceived as paragons of mental fortitude, experience mental health disorders at rates comparable to the general population. However, a pervasive culture of stigma, rooted in the narrative of "mental toughness", constitutes a formidable barrier to accessing care, leading to under-treatment and prolonged suffering. This paper addresses this critical gap by proposing *AthlyticsMind*, a novel, tailored Large Language Model (LLM)-based conversational agent designed as a confidential, stigma-free first point of contact for elite athletes. We first synthesize the epidemiological evidence on athlete mental health and deconstruct the socio-cultural mechanisms of stigma that deter help-seeking. Grounded in recent studies that critically compare the therapeutic competencies of LLMs and human psychologists, we argue that while LLMs cannot replicate the relational depth of human therapists, their strengths in procedural adherence, psychoeducation, and providing a non-judgmental interface make them ideally suited for an initial-engagement, gateway role. We present the conceptual architecture of *AthlyticsMind*, featuring sport-specific dialogue management and a curated knowledge base to address the unique stressors of an athletic career. Our primary contribution is the proposal of a hybrid human-AI model of care, where the chatbot serves as a crucial, stigma-mitigating bridge to professional support, thereby fostering a more accessible and supportive mental health ecosystem in elite sport.

## 1 Introduction

The world of elite sport is built upon a narrative of extraordinary human capacity. Athletes are celebrated for their physical prowess, discipline, and, above all, their mental toughness—an unyielding resilience in the face of immense pressure. This public image, however, belies a more complex and vulnerable reality. A growing body of evidence reveals that elite athletes are not immune to mental health challenges; in fact, they experience conditions such as anxiety and depression at rates broadly comparable to, and in some cases exceeding, those of the general population [1]. The intense mental and physical demands, coupled with unique 'workplace' stressors like relentless public scrutiny, the constant threat of career-ending injury, and the pressures of performance, create a high-risk environment [1]. This creates a fundamental paradox: the very individuals held up as symbols of psychological strength are situated in an ecosystem that can profoundly compromise their mental well-being.

Despite the clear need, a significant portion of athletes suffering from mental health symptoms do not seek professional help [2, 3]. The primary obstacle is not a lack of available services but a powerful and deeply ingrained cultural stigma [2]. The ethos of "mental toughness" functions as a double-edged sword. While essential for competition, it is often culturally misinterpreted as the stoic suppression of all vulnerability. Within this framework, acknowledging a mental health struggle

is perceived as a sign of weakness, a direct contradiction to the athlete's core identity [4]. This cultural barrier creates a "wall of silence", where athletes fear that disclosing their difficulties could jeopardize their position on a team, their selection for competition, or lucrative sponsorship deals [2]. The problem, therefore, is not merely logistical but deeply cultural - a fundamental mismatch between the constructed identity of an elite athlete and the act of seeking help.

This paper proposes a technological intervention designed to bridge this stigma-induced gap in care. We introduce *AthlyticsMind*, a specialized, confidential, and anonymous Large Language Model (LLM)-based conversational agent. Crucially, *AthlyticsMind* is not envisioned as a replacement for human therapists. Instead, it is designed to function as a critical first-line intervention—a stigma-free gateway to support. Its primary purpose is to offer immediate, non-judgmental psychoeducation and emotional support to a population culturally conditioned to avoid traditional care pathways. This proposal is grounded in a rigorous analysis of emerging research that compares the therapeutic competencies of advanced LLMs with those of human counselors [5]. By leveraging the unique strengths of AI - accessibility, anonymity, and structured interaction - we argue that a tailored chatbot can effectively lower the initial barrier to entry, creating a safe and private space for athletes to take the first step toward acknowledging their mental health needs and, ultimately, connecting with professional human care.

## 2 The Landscape of Mental Health in Elite Athletics

### 2.1 A Synthesis of Epidemiological Evidence

The notion that elite athletes are exceptionally resilient to mental illness has been systematically dismantled by recent epidemiological research. Systematic reviews and meta-analyses provide robust evidence quantifying the scale of the issue. A landmark meta-analysis by Gouttebarge et al. found that among current elite athletes, the prevalence of mental health symptoms and disorders ranged from 19% for alcohol misuse to a striking 34% for anxiety and depression [6]. This research also highlights that the period of transition out of a sporting career is one of heightened vulnerability, with former elite athletes showing prevalence rates of 16% for distress and 26% for anxiety and depression [6].

These findings are corroborated by earlier work, such as the systematic review by Rice et al., which concluded that elite athletes face a broadly comparable risk of high-prevalence mental disorders relative to the general population [1]. This body of work collectively refutes the myth of athlete immunity and establishes mental health as a significant occupational health concern within professional sports. While some studies have found that athletes may report better mental health on certain measures compared to non-athlete peers (e.g., a student population), these findings are not universal and often coexist with evidence of significant risk in other areas, underscoring the complexity of the issue [1, 2]. Table 1 provides a synthesized overview of prevalence rates for key mental health symptoms based on meta-analytic data.

Table 1: Prevalence of Key Mental Health Symptoms in Elite Athletes (A Synthesis of Meta-Analytic Data)

| Mental Health Symptom/Disorder | Prevalence in Current Athletes (%) | Prevalence in Former Athletes (%) |
|---|---|---|
| Anxiety/Depression | 34 | 26 |
| Distress | - | 16 |
| Sleep Disturbance | - | - |
| Alcohol Misuse | 19 | - |

Data synthesized primarily from the meta-analysis by Gouttebarge et al. [6]. Dashes indicate where combined prevalence data was not specifically reported for that category in the meta-analysis.

### 2.2 Analysis of Sport-Specific Stressors and At-Risk Populations

The risk of mental ill-health in athletes is not uniform; it is modulated by a range of sport-specific stressors and demographic factors. Severe musculoskeletal injuries are a potent trigger for mental

health issues, with injured athletes being at a greater risk of developing symptoms of anxiety and depression [1]. Other significant factors include performance slumps, overtraining, and burnout, which are inherent risks in a high-performance environment [7, 8].

Furthermore, the athlete population is not a monolith. Research indicates that female athletes may be more likely to report symptoms of anxiety and depression than their male counterparts [1, 3, 8]. Cultural and social factors also play a role, with evidence suggesting a lower acceptability of mental health symptoms among non-white athletes, potentially creating additional barriers to disclosure and help-seeking [2]. The type of sport may also be a factor, with some research indicating that athletes in individual sports might face different mental health challenges compared to those in team sports [8, 9, 10]. This heterogeneity is critical; it demonstrates that a one-size-fits-all approach to mental health support is destined to fail. The specific context of an athlete's career stage, injury status, gender, and sport must inform the design of any effective intervention. A generic mental health application, however well-designed for the general public, would lack the nuanced understanding required to address the specific, context-dependent stressors that define an athlete's life. This reality provides a compelling rationale for a tailored support system, one with a knowledge base and conversational capabilities specifically curated for the athletic journey.

## 3   The Stigma Barrier: A Cultural Impasse in Sport

### 3.1   Deconstructing the "Mental Toughness" Narrative

The culture of elite sport is dominated by the powerful narrative of "mental toughness". While this trait is undeniably crucial for high performance, its cultural definition often becomes conflated with an unyielding stoicism and the complete suppression of emotional vulnerability [3, 11]. In this environment, mental illness is not seen as a medical condition but as a character flaw - a sign of being "mentally weak" [3]. This framing creates a profound cultural impasse. Athletes are caught in a double bind: they are expected to be open about physical injuries, for which a team of trainers and physicians stands ready to assist, but are implicitly discouraged from disclosing psychological struggles [3]. The fear of being judged by coaches, teammates, and fans - and the potential for negative career repercussions, such as being dropped from a team or losing a contract - becomes a powerful deterrent to seeking help [2].

### 3.2   Mechanisms of Stigma and Impression Management

This cultural pressure forces athletes to engage in sophisticated forms of "impression management" to protect their social and professional standing [4]. Research has identified several specific strategies that athletes use to conceal their psychological suffering. These include "wearing a mask" to project an image of strength and invulnerability, adhering to a "vow of silence" that renders stories of mental struggle untellable within the sport environment, and finding "alibis" by attributing psychological symptoms to more acceptable physical causes [4].

These behaviors are driven by two distinct but related forms of stigma. The first is *perceived public stigma* - the athlete's belief that others (coaches, teammates, the public) hold negative views about mental illness and that disclosure will lead to negative consequences [3]. The second is *self-stigma*, where the athlete internalizes these negative societal beliefs, leading to feelings of shame and embarrassment that prevent them from even acknowledging their own struggles [3]. This dynamic can be visualized as a self-reinforcing cycle (Figure 1), where performance pressure fosters a culture that stigmatizes vulnerability, leading to concealment, which in turn prevents help-seeking, exacerbates symptoms, and ultimately impairs performance, thus intensifying the initial pressure.

This understanding of the sporting world as a "front stage" - a term from sociology where individuals are constantly performing for an audience - is crucial. The locker room, the training ground, and the media interview are all stages where the mask of the invulnerable athlete must be worn. What is conspicuously absent is a safe "backstage", a private space where this performance can be dropped and authentic vulnerability can be expressed without fear of reprisal. A confidential, anonymous, and non-judgmental chatbot can be conceptualized as precisely this: a digital backstage. Its value lies not just in its function as a tool for information delivery, but in its fundamental nature as a structurally safe space that provides a function currently missing from the elite sport ecosystem.

Figure 1: A Conceptual Model of Stigma's Cycle in Elite Sport.

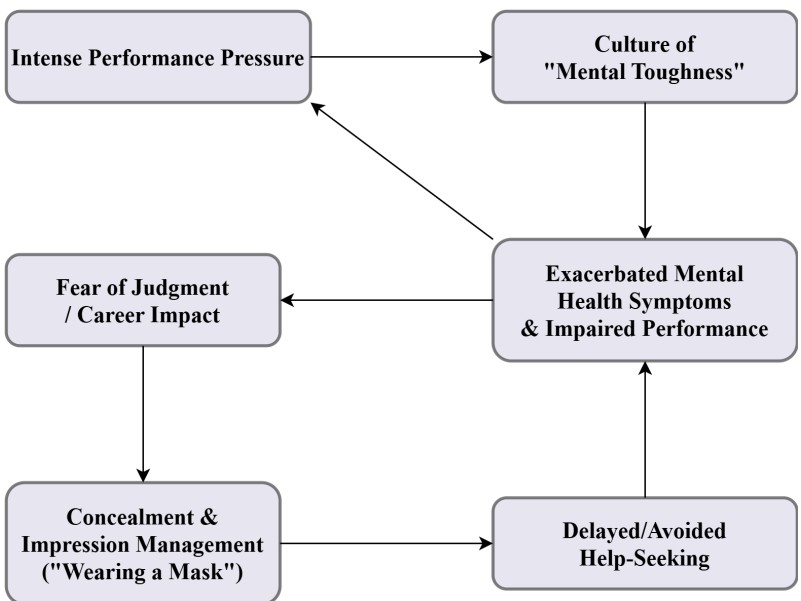

This model illustrates the reinforcing feedback loop where performance pressure and the culture of mental toughness lead to stigma, concealment, and avoidance of care, ultimately exacerbating the initial problem.

## 4 Conversational AI as a Stigma-Free Support Modality

### 4.1 The Potential of LLMs to Bypass Traditional Barriers

Large Language Models (LLMs) offer a promising modality for mental health support precisely because their core attributes directly counteract the primary barriers erected by stigma. Conversational agents powered by LLMs can provide support that is available 24/7, completely anonymous, and inherently non-judgmental [12]. This creates an accessible and de-stigmatized channel for eHealth services, allowing individuals to seek initial support without the fear of social or professional repercussions [12]. For an athlete hesitant to speak to a coach or team psychologist, the ability to interact with a confidential system at any time, from any location, represents a profound shift in the accessibility of care.

### 4.2 A Critical Review of LLM vs. Human Therapeutic Competencies

The proposal to use an LLM for mental health support must be grounded in a clear-eyed assessment of its capabilities and limitations relative to human practitioners. The debate should not be framed as a simple competition of "AI versus human", but as an analysis of complementary strengths. A recent mixed-methods study by Iftikhar et al. provides a detailed comparison of LLM-based counselors and human peer counselors in delivering single-session Cognitive Behavioral Therapy (CBT), revealing a distinct functional trade-off [5].

LLMs demonstrate notable strengths in procedural and structural aspects of therapy. They exhibit higher adherence to the specific protocols of a therapeutic modality like CBT, are more effective at setting and maintaining a session agenda, and excel at delivering structured psychoeducation [5]. In contrast, human counselors excel at relational strategies. They are superior at building a strong therapeutic alliance through genuine empathy, appropriate self-disclosure, and a nuanced understanding of cultural and personal context. This relational skill fosters deeper collaboration and user self-reflection [5].

However, both have weaknesses. The LLM's rigid adherence to protocol can come at the expense of collaboration; it may misread subtle social cues, produce "deceptive empathy"—a formulaic warmth

that lacks genuine connection—and treat the user as a passive recipient of information rather than an active partner in a therapeutic dialogue [5]. Human counselors, while relationally adept, can be less structured, deviate from the therapeutic agenda, and may not adhere as strictly to a given protocol [5]. This comparative analysis is summarized in Table 2.

Table 2: Comparative Analysis of Human vs. LLM Therapeutic Competencies in Single-Session CBT

| Therapeutic Dimension | Human Peer Counselor | LLM-based Counselor |
|---|---|---|
| **Therapeutic Alliance** (Empathy, Collaboration) | **Strengths:** Excels at relational strategies (small talk, self-disclosure) to build rapport. Higher demonstrated empathy and collaboration, leading to deeper user reflection. Adept at picking up implicit cues. | **Weaknesses:** Can produce "deceptive empathy" (formulaic warmth). Struggles to sustain collaboration and can be cold or formulaic. Lacks genuine interpersonal effectiveness. |
| **Adherence to Method** (Structure, Psychoeducation) | **Weaknesses:** Sessions can be less structured and may deviate from the agenda. Can engage in non-therapeutic digressions. | **Strengths:** Demonstrates higher procedural adherence to CBT techniques. Sessions are more structured with clear agenda-setting. Excels at delivering psychoeducation. |
| **Guided Discovery** (User Reflection) | **Strengths:** Fosters active user participation and exploration through skillful, open-ended questioning. Guides users to their own conclusions. | **Weaknesses:** Tends to treat users as passive participants, "lecturing" or spoon-feeding information. Undermines collaborative discovery by imposing solutions. |

This table synthesizes findings from Iftikhar et al. [5], highlighting the distinct and complementary strengths and weaknesses of human and AI-based counselors.

## 4.3 Argument for a Gateway Model

This evidence of functional divergence suggests that the most effective application of LLMs in mental healthcare is not as a standalone replacement for therapists, but as a component in a carefully designed workflow. The LLM's strengths - anonymity, structure, and reliable delivery of psychoeducation - are perfectly aligned with the needs of an individual at the very beginning of their help-seeking journey, particularly one constrained by stigma. An athlete's primary needs at this stage are not deep relational therapy but a safe, private, and informative first step. The LLM can serve as an ideal "gateway": a tool for initial engagement, symptom checking, and mental health literacy building. It can triage needs and provide foundational support, functioning as a precursor to human care rather than a competitor. This reframes the entire paradigm from "human vs. AI" to a "human-AI workflow", where technology is used to overcome the initial barriers and facilitate a "warm handoff" to the nuanced, empathetic care that only a human professional can provide.

# 5 Proposed System: The AthlyticsMind Chatbot Architecture

## 5.1 Conceptual Framework

Building on the identified needs of elite athletes and the specific capabilities of LLMs, we propose *AthlyticsMind*, a chatbot architected specifically for this population. The system is founded on several core design principles:

- **Unyielding Confidentiality:** All interactions must be anonymous and protected by end-to-end encryption to build the trust necessary to overcome stigma.

- **Athlete-Centric Content:** The language, examples, and knowledge base must be tailored to the world of elite sport, using terminology and addressing scenarios that are immediately recognizable and relevant to the user.

- **Flexible and Accessible:** The system must be available on-demand, fitting into the demanding and often unpredictable schedules of athletes.

- **Safety-First Escalation:** A clear and robust protocol for detecting and responding to user crises is a non-negotiable safety feature.

## 5.2 Core Architectural Modules

The proposed architecture for *AthlyticsMind* integrates a state-of-the-art LLM with several specialized components designed to meet the unique needs of elite athletes. This design moves beyond a generic mental health chatbot to create a tailored, context-aware system. The architecture, depicted in Figure 2, consists of six core modules. This design demonstrates a proactive stance on responsible AI; ethical considerations are not an afterthought but are embedded directly into the system's structure, particularly through the security and crisis detection modules.

1. **Secure User Interface:** A mobile and web-based front-end that serves as the user's point of interaction. It is designed for simplicity and, most critically, ensures user anonymity and end-to-end encryption for all communications.

2. **Natural Language Understanding (NLU) Engine:** This module utilizes a powerful pre-trained LLM (e.g., from the GPT or Llama families) to parse and interpret the user's natural language input, identifying intent, entities, and sentiment.

3. **Sport-Specific Dialogue Manager:** This is the central processing unit of the system. Unlike a generic dialogue manager, this module is fine-tuned on data and rules relevant to the athletic experience. It manages the conversational flow, tracks the user's state (e.g., discussing performance anxiety vs. injury recovery), and selects appropriate response strategies from a predefined playbook.

4. **Athlete-Centric Knowledge Base:** A curated and verified database of information that the Dialogue Manager can query. This knowledge base contains content specifically developed for athletes on topics such as managing performance anxiety, coping with public scrutiny, the psychology of injury rehabilitation, sleep hygiene for optimal recovery, and navigating career transitions [2, 4, 6].

5. **Response Generation Engine:** This module uses the core LLM, guided by instructions from the Dialogue Manager, to generate responses. The prompts are engineered to produce replies that are empathetic, supportive, and aligned with evidence-based psychoeducational principles, using language and analogies that resonate with an athletic mindset.

6. **Crisis Detection & Escalation Protocol:** A critical safety layer that operates in parallel. It uses a combination of keyword detection (e.g., "suicide", "self-harm") and sentiment analysis to monitor for signs of acute distress [13, 14, 15]. If a predefined risk threshold is crossed, the system interrupts the standard conversational flow to immediately provide the user with localized, actionable crisis resources (e.g., national hotlines, contact information for the user's team psychologist if available) and offers a pathway to connect with a human counselor.

# 6 Discussion and Future Directions

## 6.1 Implications for a Hybrid Model of Care and Scientific Discovery

The true potential of *AthlyticsMind* is realized when it is viewed not as an isolated tool, but as a foundational component of a broader, integrated ecosystem. In its clinical role, the chatbot serves as the initial, low-threshold entry point for a hybrid model of care. It can handle the crucial early stages of engagement: building mental health literacy, normalizing the experience of psychological distress, and offering basic coping strategies. This function can significantly reduce the burden on human practitioners, allowing them to dedicate their time to athletes who are ready for deeper therapeutic work. Furthermore, the system could be designed to facilitate a "warm handoff". With explicit user

Figure 2: Architectural Diagram of the AthlyticsMind Chatbot.

This diagram illustrates the flow of information within the *AthlyticsMind* system. User input is processed by the NLU Engine and passed to the specialized Dialogue Manager, which leverages a curated Knowledge Base and a Crisis Detection module to instruct the LLM Core in generating a safe and relevant response.

consent, the chatbot could generate a confidential summary of conversations, which the athlete could then choose to share with a human therapist, bridging the gap between the digital and clinical realms.

Beyond this clinical utility, this model has profound implications for research, positioning *Athlytics-Mind* as a novel scientific instrument for sports psychology. By aggregating anonymized interaction data, the system can provide unprecedented, real-time insights into the mental health of a tradition-ally hard-to-study population. This reframes the agent as a discovery engine, capable of helping researchers answer critical scientific questions:

- How do self-reported mental health symptoms fluctuate in relation to competition schedules, injuries, or media events?

- What conversational pathways and psychoeducational strategies are most correlated with an increase in a user's intent to seek human-led care?

- Can we identify emergent, sport-specific stressors from the anonymized conversational topics that are not yet prominent in the literature?

This data-driven approach allows the system to not only provide support but also to contribute to a deeper, evolving understanding of athlete mental well-being, creating a virtuous cycle where research insights are used to continually improve the intervention itself.

## 6.2 Addressing Critical Limitations and Ethical Imperatives

The deployment of an LLM in such a sensitive domain necessitates a frank acknowledgment of its limitations and a commitment to rigorous ethical oversight. Several challenges are paramount:

- **Data Privacy and Security:** The protection of highly sensitive personal health information is non-negotiable. The system must adhere to the highest standards of data encryption, anonymization, and security protocols to prevent any potential for data leakage or misuse [12].

- **Algorithmic Bias:** LLMs are trained on vast datasets from the general internet, which may contain biases. There is a risk that the model could fail to respond sensitively to the unique experiences of athletes from diverse racial, cultural, or socioeconomic backgrounds. Ongoing auditing and fine-tuning with representative data are essential to mitigate this risk [12].

- **"Deceptive Empathy" and Over-Reliance:** As identified in comparative studies, LLMs can produce formulaic responses that mimic empathy without genuine understanding [5]. It is ethically imperative that the system is transparent about its nature as an AI. Users must not be misled into forming deep parasocial relationships or becoming overly reliant on the chatbot, which could deter them from seeking necessary human connection [12].

- **Crisis Management Limitations:** While the proposed architecture includes a crisis detection protocol, no automated system can be foolproof in identifying and managing severe mental health crises. The system must be clear about its limitations and its primary role as a supportive tool, not a crisis intervention service.

### 6.3 A Research Roadmap for Validation

Moving *AthlyticsMind* from a conceptual architecture to a validated clinical tool requires a phased research program. The initial phase would involve prototype development and qualitative pilot studies with a small cohort of current and former elite athletes. The goal of this phase would be to assess usability, acceptability, and the relevance of the content within the knowledge base. Feedback from these studies would inform iterative design improvements.

The second phase would consist of a larger-scale randomized controlled trial (RCT). In this trial, athletes would be randomized to one of three groups: a group with access to *AthlyticsMind*, a group provided with a list of traditional mental health resources, and a waitlist control group. The primary outcomes would be measures of mental health literacy, help-seeking intentions, and stigma reduction. Secondary outcomes would include metrics of engagement with the chatbot and, crucially, rates of successful connection to human-provided mental health services. Such a trial would provide the rigorous empirical evidence needed to validate the efficacy of the chatbot as a gateway to care.

## 7 Conclusion

The mental health of elite athletes is a significant and complex issue, compounded by a deeply rooted culture of stigma that often prevents those in need from seeking help. While the challenge is formidable, advancements in artificial intelligence offer a novel and promising path forward. This paper has argued that a tailored Large Language Model-based conversational agent, *AthlyticsMind*, can serve as a powerful tool to dismantle these barriers. By providing a confidential, accessible, and non-judgmental first point of contact, such a system can function as a crucial gateway to the broader ecosystem of mental healthcare. Grounded in evidence that highlights the complementary strengths of AI and human counselors, we propose a hybrid model of care where technology is not a replacement for human connection but a bridge to it. Through careful design, ethical oversight, and rigorous validation, tools like *AthlyticsMind* have the potential to foster a healthier, more supportive, and more open sporting environment for all athletes.

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

## Agents4Science AI Involvement Checklist

1. **Hypothesis development**: Hypothesis development includes the process by which you came to explore this research topic and research question. This can involve the background research performed by either researchers or by AI. This can also involve whether the idea was proposed by researchers or by AI.

   Answer: [C]

   Explanation: The core research hypothesis—that a tailored LLM agent can act as a stigma-mitigating gateway for athlete mental health—was co-developed. A human researcher

provided the initial high-level problem domain. The AI system then synthesized a wide range of literature from sports psychology, HCI, and AI ethics to formulate the specific, testable hypothesis and the novel "gateway" model concept. The AI's contribution was central to refining the initial idea into a structured research question and framework.

2. **Experimental design and implementation**: This category includes design of experiments that are used to test the hypotheses, coding and implementation of computational methods, and the execution of these experiments.

Answer: [C]

Explanation: As this is a conceptual paper, there was no physical implementation. However, the conceptual architecture, the definition of core modules (e.g., Sport-Specific Dialogue Manager, Crisis Detection Protocol), and the proposed research roadmap for validation (including the RCT design) were primarily generated by the AI based on best practices in software engineering and clinical trial design. Human researchers provided feedback and refinement, but the detailed design was majority AI-driven.

3. **Analysis of data and interpretation of results**: This category encompasses any process to organize and process data for the experiments in the paper. It also includes interpretations of the results of the study.

Answer: [C]

Explanation: This paper does not contain experimental data. The "analysis" consisted of a large-scale synthesis of existing literature. The AI performed the majority of this work, identifying, summarizing, and critically analyzing key papers (e.g., Iftikhar et al., Gouttebarge et al.) to build the logical argument for the proposed system. Human oversight ensured the interpretation was accurate and relevant, but the AI performed the initial analytical heavy lifting to draw connections between disparate fields.

4. **Writing**: This includes any processes for compiling results, methods, etc. into the final paper form. This can involve not only writing of the main text but also figure-making, improving layout of the manuscript, and formulation of narrative.

Answer: [C]

Explanation: The AI generated the initial draft of the entire manuscript, including the abstract, introduction, all sections, and the initial LaTeX formatting. The narrative structure, such as framing the problem as a "wall of silence" and the solution as a "gateway," was proposed and written by the AI. Human researchers performed a crucial role in editing, fact-checking, refining the prose for clarity and tone, and correcting formatting errors, but the bulk of the text (>50%) was AI-generated.

5. **Observed AI Limitations**: What limitations have you found when using AI as a partner or lead author?

Description: While the AI demonstrated strong capabilities in literature synthesis and drafting, we observed several limitations. The primary issue was a tendency toward "hallucination" or confident inaccuracy, particularly with bibliographic details and citations, which required meticulous human fact-checking. The AI sometimes produced overly formulaic or generic text that lacked a strong, persuasive voice, requiring human intervention to refine the narrative. Finally, without explicit guidance, the AI struggled to perfectly align the paper's framing with the specific, niche focus of the target conference, necessitating a human-led strategic revision.

