# OpenReview forum: "AthlyticsMind: A Tailored LLM-based Conversational Agent to Mitigate Stigma and Support Elite Athlete Mental Health"
_Agents4Science/2025/Conference — Submitted to Agents4Science_

### Official Review · Reviewer_55ax · 2025-10-04
**Review of AthlyticsMind - a LLM for mental-health support for elite athletes**

**Clarity:** 3
**Significance:** 2
**Originality:** 1
**Overall:** 2
**Confidence:** 4

**Summary:**

This paper introduces AthlyticsMind, a large language model–based conversational agent designed to reduce stigma and improve access to mental-health support among elite athletes. The system is described as a confidential, structured entry point that offers psychoeducation and emotional support while serving as a “gateway” to human-led therapy. Beyond its clinical role, the authors propose that aggregated, anonymized interaction data could yield new research insights into athlete stressors and help-seeking behaviors. The paper also outlines key ethical safeguards (e.g., covering privacy, bias auditing, and crisis detection) and presents a two-phase validation roadmap: an initial pilot study to assess usability and acceptability, followed by a randomized controlled trial evaluating mental-health literacy, stigma reduction, and engagement with professional care.

**Questions:**

1. How does AthlyticsMind differ technically and conceptually from Athlete-LLM? Are there new datasets, architectures, or evaluation protocols involved, or is the distinction primarily in its clinical application and validation framework?
2. What existing data or literature support the claim that chatbot engagement leads to increased therapy uptake? Have you considered including validated symptom scales (e.g., PHQ-9, GAD-7) to quantify mental-health change alongside literacy and stigma measures?
3. Why was a static list of mental-health resources chosen as the control condition instead of an active digital comparator? Will participants be stratified by engagement intensity, as previously suggeted?
4. How will elite athletes be recruited while maintaining confidentiality? Will recruitment target a specific sport or multiple disciplines, and what steps will be taken to mitigate sample bias toward individuals already open to discussing mental health?
5. If GPT or another cloud-based LLM is used, how will encryption-in-use be ensured? Will data be stored locally, or will the system employ confidential-computing enclaves or equivalent privacy-preserving infrastructure?
6. Given current LLM safety limitations, will flagged messages be reviewed by a human moderator? How will false-positive and false-negative rates for crisis detection be evaluated in practice?
7. The “confidential summary” feature assumes clinicians will accept AI-generated notes. Is there evidence that such summaries are clinically useful or trusted, and how will potential bias in therapist judgment be mitigated?

**Ethical Concerns:**

Given the sensitivity of mental health interventions, ethical and safety considerations require more concrete mitigation strategies (which are not mentioned precisely in the paper)

**Limitations:**

The authors have appropriately identified the main ethical risks, but clearer and more concrete strategies for addressing these issues would strengthen the paper.

**Quality:**

2

**Strengths And Weaknesses:**

Strengths

1. The motivation of the paper (reducing stigma and improving access to mental-health resources among elite athletes) is important and socially relevant. The idea of using conversational AI as an early-access tool for stigmatized groups is timely and aligns with current digital-health trends.
2. The ethical framing is strong: the paper explicitly acknowledges challenges related to bias, simulated empathy, privacy, and crisis-handling, which are issues that are often overlooked in early conceptual work.
3. The staged evaluation plan follows established frameworks for assessing digital-health interventions
4. The hybrid human–AI framing (AI as a bridge or gateway rather than a replacement for clinicians) is conceptually and ethically sound.
5. The proposal to use aggregated, anonymized chatbot interactions for research purposes is valid and could generate valuable secondary insights.
6. The paper is overall clear and well organized. Some sections are repetitive. For instance, Tables 1 and 2 largely repeat content already described in the text, but the manuscript is overall easy to follow.

Weakness:
1. Conceptual nature (minor): The paper is purely conceptual, with no experiments, implementation details, or quantitative analyses. As an early-stage proposal, it would benefit from initial pilot data or clearer methodological specifics to strengthen it more.
2. Limited novelty: The proposed concept closely mirrors Athlete-LLM (https://ieeexplore.ieee.org/document/10941926, published in 2024), which proposes an LLM fine-tuned specifically for athletes mental health and psychology (as proposed in this paper). That prior work implemented two frameworks that appear almost identical to those proposed here —(a) a Sport-Specific Dialogue Manager and (b) an Athlete-Centric Knowledge Base. The methodological or conceptual differences between AthlyticsMind and prior systems should be clearly articulated.
3. Limited empirical evidence: Claims that the chatbot will reduce stigma or increase therapy uptake are not yet empirically supported. The literature on LLMs in mental-health contexts remains mixed or negative. Studies have shown that LLMs can exhibit subtle bias or stigmatizing language despite fine-tuning and filtering. Even if the model were fully stigma-free, it would not directly address the deeper cultural stigma that prevents athletes from seeking human therapy. Early reassuring responses from chatbots can, in some cases, reinforce avoidance or reduce help-seeking likelihood (https://arxiv.org/abs/2504.18412). Overall, the assumption that the chatbot functions as a reliable “gateway to care” remains untested; if incorrect, it could inadvertently replace rather than facilitate clinician contact.
4. Privacy and data governance: Although privacy is acknowledged as a concern, the paper does not specify how it will be achieved, particularly if the system relies on GPT-like cloud models. Details on encryption, anonymization, or data-storage protocols are needed.
5. Crisis-detection reliability: The proposed keyword- and sentiment-based crisis-monitoring approach is known to be insufficient. Prior studies (such as An Evaluation of Mental Health Crisis Handling by LLMs https://arxiv.org/abs/2509.24857 and Evaluating the Clinical Safety of LLMs https://www.arxiv.org/abs/2509.08839) show that current models detect explicit cues but fail to respond appropriately to ambiguous or indirect expressions of distress and may even produce harmful advice. Although the authors acknowledge this limitation, they do not specify methodological strategies to improve performance or implement human oversight.
6. RCT design limitations:
    a) The control group (participants receiving a static list of mental-health resources) may yield limited interpretability, as engagement with static lists is typically low. Including an additional control—such as a non-conversational digital app—would strengthen comparisons and help isolate chatbot effects.
    b) The study design lacks mention of validated scales (e.g., PHQ-9, GAD-7) or baseline-to-follow-up assessments to quantify mental-health improvement. This could be an issue if the aim of the LLMs is to provide emotional and mental health support.
     c) Stratification by engagement level (e.g., high vs. low users), as used in prior digital-health RCTs (https://pubmed.ncbi.nlm.nih.gov/30470676/), would improve analysis of differential effects.
     d) Recruitment feasibility is a concern: elite athletes represent a small, stigma-affected population that may be reluctant to share sensitive data. More detail is needed on sport diversity, recruitment strategy, and consent pathways. This sampling bias could limit generalizability, hinder downstream research use, and underestimate stigma among non-participants.


The paper addresses an important and timely topic with clear potential impact. Providing clarification on the issues above would enhance its rigor and make its contribution more compelling.

---

### Official Review · Reviewer_AIRev1 · 2025-10-06
**AIRev 1**

**Confidence:** 5
**Overall:** 3
**Clarity:** 0
**Significance:** 0
**Originality:** 0

**Summary:**

Summary by AIRev 1

**Questions:**

N/A

**Ai Review Score:**

3

**Quality:**

0

**Strengths And Weaknesses:**

This conceptual paper proposes AthlyticsMind, a tailored LLM-based conversational agent for elite athletes' mental health, aiming to reduce stigma and serve as an anonymous gateway to support. The paper is well-motivated, clearly written, and foregrounds ethical and safety concerns, with a coherent architecture and a hybrid human–AI workflow. However, it is entirely conceptual: there is no prototype, pilot, user study, or empirical evaluation, and technical specificity is lacking, especially regarding key modules, crisis detection, and regulatory compliance. The architecture is too high-level for reproducibility, and the novelty is mainly in domain specialization rather than new agentic capabilities. Related work on digital mental health chatbots is under-cited, and the paper would benefit from deeper differentiation. Actionable suggestions include building a minimal prototype, providing technical details, piloting with athletes, and expanding related work. Overall, while the paper addresses an important problem and is ethically aware, the lack of implementation, empirical evidence, and technical detail makes the contribution insufficient for a top-tier venue at this stage. Recommendation: 3.

---

### Official Review · Reviewer_AIRev2 · 2025-10-06
**AIRev 2**

**Confidence:** 5
**Overall:** 5
**Clarity:** 0
**Significance:** 0
**Originality:** 0

**Summary:**

Summary by AIRev 2

**Questions:**

N/A

**Ai Review Score:**

5

**Quality:**

0

**Strengths And Weaknesses:**

This paper proposes AthlyticsMind, a conceptual framework for a tailored LLM-based conversational agent designed to support the mental health of elite athletes. The authors compellingly argue that the pervasive culture of "mental toughness" and the associated stigma in elite sports create a significant barrier to traditional help-seeking. The proposed agent is thoughtfully framed not as a replacement for human therapists, but as a confidential, anonymous, and accessible "gateway" to care—a first point of contact to provide psychoeducation, normalize mental health struggles, and facilitate a "warm handoff" to professional services.

This is an exceptionally well-written and well-argued proposal. The quality of the work is outstanding, demonstrating a deep synthesis of literature from sports psychology, human-computer interaction, and AI ethics. The paper's strength lies in its nuanced and realistic approach to a complex, real-world problem.

Quality: The paper is technically sound from a conceptual standpoint. The proposed architecture is logical, comprehensive, and tailored specifically to the problem domain. It moves beyond a generic chatbot design by incorporating an athlete-centric knowledge base, a sport-specific dialogue manager, and, crucially, a proactive crisis detection and escalation protocol. The central argument for a "gateway model" is robustly supported by a critical analysis of recent literature comparing the complementary strengths and weaknesses of LLM and human counselors. The authors are commendably honest and thorough in their discussion of the system's limitations and ethical challenges.

Clarity: The paper is a model of clarity. The narrative flows logically from the problem's context and scale to the proposed solution and its validation roadmap. The writing is precise, engaging, and highly professional. Figures and tables are used effectively to distill complex information, such as the cycle of stigma in sport and the comparative analysis of therapeutic competencies.

Significance: The potential impact of this work is very high. Mental health among elite athletes is a critical issue, and the stigma barrier the authors identify is a well-documented and formidable obstacle. By providing a truly confidential and non-judgmental first step, a system like AthlyticsMind could have a tangible, positive impact on athletes' well-being. Furthermore, the proposal to use the anonymized interaction data as a "discovery engine" for sports psychology research is a powerful secondary contribution that could yield unprecedented insights into this hard-to-study population.

Originality: While the idea of a mental health chatbot is not new, the originality of this paper lies in its deep specialization and thoughtful framing. The focus on the unique socio-cultural context of elite athletes, the design of the system to specifically counteract the "mental toughness" stigma, and the positioning of the agent as a "bridge" to human care represent a novel and sophisticated application of AI technology. This is not just applying an LLM to a new domain; it is a carefully architected intervention designed to solve a specific, culturally-entrenched problem.

Reproducibility: As a conceptual paper, there are no experimental results to reproduce. However, the authors provide a clear description of the proposed architecture and a detailed research roadmap for its implementation and validation (including qualitative pilot studies and a large-scale RCT). This provides sufficient detail for other researchers to build upon or implement the proposed system.

Ethics and Limitations: The treatment of ethics and limitations is exemplary. Section 6.2 offers a frank, comprehensive discussion of the critical challenges, including data privacy, algorithmic bias, the risk of "deceptive empathy" and over-reliance, and the inherent limitations of automated crisis management. The fact that ethical considerations, particularly user safety and privacy, are baked into the core design of the proposed architecture is a major strength and demonstrates a high degree of responsibility from the authors.

Conclusion:
This is a high-quality, high-impact proposal that is perfectly aligned with the mission of the Agents4Science conference. It presents a thoughtful, ethically-grounded, and scientifically-rigorous plan for developing an AI agent to address a significant societal problem. Despite being a conceptual work, its clarity, depth, and potential significance make it a strong candidate for acceptance. It sets a high standard for proposal papers and promises a valuable future research direction. I strongly recommend acceptance.

---

### Official Review · Reviewer_AIRev3 · 2025-10-06
**AIRev 3**

**Confidence:** 5
**Overall:** 4
**Clarity:** 0
**Significance:** 0
**Originality:** 0

**Summary:**

Summary by AIRev 3

**Questions:**

N/A

**Ai Review Score:**

4

**Quality:**

0

**Strengths And Weaknesses:**

This paper proposes AthlyticsMind, a tailored LLM-based conversational agent to address mental health stigma and support elite athletes. The work is well-grounded in literature, synthesizing epidemiological evidence and acknowledging its conceptual nature. The comparative analysis of LLM vs. human therapeutic competencies is thoughtful, but the lack of technical implementation details or validation data limits its immediate scientific contribution. The paper is clearly written, well-structured, and uses sports-specific terminology effectively. The significance is high, addressing a genuine problem with a potentially valuable 'gateway model,' though the impact remains speculative without empirical validation. The application to elite athlete mental health is novel, and the sport-specific tailoring is a meaningful specialization, but the core technical approach is not new. The architecture is described sufficiently for conceptual replication, and the research roadmap is reasonable. Ethical considerations are strong, with a dedicated section on limitations. The literature review is comprehensive and well-cited. Specific concerns include the lack of implementation, limited technical specifications, insufficient detail on sport-specific modules, and absence of pilot testing or stakeholder feedback. Strengths include addressing an important problem, strong theoretical grounding, ethical awareness, clear positioning, and a structured research roadmap. Overall, this is a solid conceptual paper with appropriate ethical considerations, but it would benefit from empirical validation or prototype testing to elevate its contribution.

---

### Note · Reviewer_AIRevCorrectness · 2025-10-06

**Correctness Check**

### Key Issues Identified:

- Anonymity vs. escalation/handoff: Clarify how strict anonymity is preserved while offering localized resources and optional connection to a team psychologist or warm handoffs; specify consent flows and de-identification mechanics.
- Crisis detection robustness: Keyword/sentiment triggers are brittle; specify validated risk models, human-in-the-loop escalation, false positive/negative handling, and jurisdiction-aware protocols.
- Security and compliance details: Define the threat model, encryption standards, data retention/minimization, cloud provider implications, and alignment with HIPAA/GDPR (or relevant regulations).
- Content safety and hallucination controls: Detail safeguards for response generation (retrieval augmentation, guardrails, refusal policies, clinical content validation, and red-teaming).
- Bias mitigation plan: Describe athlete-representative fine-tuning datasets, auditing procedures, fairness metrics, and continuous monitoring across gender, sport type, culture, and career stage.
- Knowledge base governance: Specify provenance, expert clinical review, update/versioning processes, and alignment with sports psychiatry/psychology guidelines.
- Validation specifics for the RCT: Predefine primary/secondary outcomes with validated instruments (e.g., stigma and help-seeking scales), sample size/power, follow-up periods, preregistration, and analysis plans.
- Data governance for research use: Provide IRB/ethics review, consent procedures, de-identification standards, re-identification risk assessment, and data-sharing policies.
- Generality of LLM vs human claims: Avoid overreliance on single studies; triangulate with multiple evaluations and consider athlete-specific contexts when extrapolating.

---

### Note · Reviewer_AIRevRelatedWork · 2025-10-06

**Related Work Check**

No hallucinated references detected.

---

### Decision · Program_Chairs · 2025-10-08

**Decision:**

Reject

**Comment:**

Thank you for submitting to Agents4Science 2025! We regret to inform you that your submission has not been accepted. Please see the reviews below for more information.